# Care for critically and terminally ill patients and moral distress of physicians and nurses in tertiary hospitals in South Korea: A qualitative study

**Jiyeon Kang**[1☯], **Eun Kyung Choi**[2☯], **Minjeong Seo**[3], **Grace S. Ahn**[4], **Hye Youn Park**[5], **Jinui Hong**[6], **Min Sun Kim**[6,7], **Bhumsuk Keam**[6,8], **Hye Yoon Park**[6,9,10]*

1 Department of Anthropology, Seoul National University, Gwanak-gu, Seoul, Republic of Korea, 2 Medical Education Center, School of Medicine, Kyungpook National University, Daegu, Republic of Korea, 3 College of Nursing and Gerontological Health Research Center in Institute of Health Sciences, Gyeongsang National University, Jinju, Gyeongsangnamdo, Republic of Korea, 4 School of Medicine, University of California San Diego, La Jolla, CA, United States of America, 5 Department of Neuropsychiatry, Seoul National University Bundang Hospital, Seongnam, Korea, 6 Center for Palliative Care and Clinical Ethics, Seoul National University Hospital, Jongno-gu, Seoul, Republic of Korea, 7 Department of Pediatrics, Seoul National University Hospital, Jongno-gu, Seoul, Republic of Korea, 8 Department of Internal Medicine, Seoul National University Hospital, Jongno-gu, Seoul, Republic of Korea, 9 Department of Psychiatry, Seoul National University Hospital, Jongno-gu, Seoul, Republic of Korea, 10 Department of Psychiatry, Seoul National University, College of Medicine, Jongno-gu, Seoul, Republic of Korea

☯ These authors contributed equally to this work.
* psychepark@gmail.com

**Data Availability Statement:** This study contains ethically sensitive data, which may endanger the participants of this study. If a researcher or a

## Abstract

Physicians and nurses working in acute care settings, such as tertiary hospitals, are involved in various stages of critical and terminal care, ranging from diagnosis of life-threatening diseases to care for the dying. It is well known that critical and terminal care causes moral distress to healthcare professionals. This study aimed to explore moral distress in critical and terminal care in acute hospital settings by analyzing the experiences of physicians and nurses from various departments. Semi-structured in-depth interviews were conducted in two tertiary hospitals in South Korea. The collected data were analyzed using grounded theory. A total of 22 physicians and nurses who had experienced moral difficulties regarding critical and terminal care were recruited via purposive maximum variation sampling, and 21 reported moral distress. The following points were what participants believed to be right for the patients: minimizing meaningless interventions during the terminal stage, letting patients know of their poor prognosis, saving lives, offering palliative care, and providing care with compassion. However, family dominance, hierarchy, the clinical culture of avoiding the discussion of death, lack of support for the surviving patients, and intensive workload challenged what the participants were pursuing and frustrated them. As a result, the participants experienced stress, lack of enthusiasm, guilt, depression, and skepticism. This study revealed that healthcare professionals working in tertiary hospitals in South Korea experienced moral distress when taking care of critically and terminally ill patients, in similar ways to the medical staff working in other settings. On the other hand, the present study uniquely identified that the aspects of saving lives and the necessity of palliative care were reported

research team, with a clear research object, specific methods and ethical approval to review this data, requests access to this data, please contact SNUH IRB (snuhirb@gmail.com, +82-2-2072-1601). The result of research utilizing this data may be reviewed by the authors before publication.

**Funding:** HYP received a research fund from Seoul National University Hospital (no. 04-2016-0660). The funders had no role in study design, data collection and analysis, decision to publish, or preparation of the manuscript.

**Competing interests:** The authors have declared that no competing interests exist.

as those valued by healthcare professionals. This study contributes to the literature by adding data collected from two tertiary hospitals in South Korea.

## Introduction

Tertiary hospitals constitute a starting point of care for critically and terminally ill patients. Patients visit tertiary hospitals to seek treatment, cure, and recovery, but some patients with life-threatening diseases are categorized as being under life-or-death emergency situations. Some patients struggling with critical diseases are diagnosed to be at the terminal stage, while some at the stage die in the hospitals. Recent statistics support the current worldwide status of hospitals as places for dying. Half of the people in the United Kingdom (UK) and Australia die in hospitals [1]. Compared to some countries, such as the United States of America (USA), where the percentage of deaths that occurred in hospitals has decreased from 48.0% in 2000 to 35.1% in 2018 [2], in East Asian countries, the proportion of hospital deaths is more salient. In 2018, nearly 90% of the total deaths in Hong Kong, 80% in Japan, and 62.5% in Singapore occurred in hospitals [3]. Similarly, in South Korea, 76.2% of all deaths in 2017 occurred in hospitals [4].

Despite the large proportion of hospital deaths, only few tertiary hospitals in South Korea provide palliative and hospice care services for their outpatients and inpatients. For example, only 16 tertiary hospitals among 45 were equipped with palliative and hospice care wards or teams in 2017. The main goal of tertiary hospitals when dealing with terminal patients is to refer them to hospices or nursing homes; however, due to the insufficient number of hospices, physicians and nurses often fail to do so. [5]. In addition, until the introduction of advance directives (AD) and physician order for life-sustaining treatment (POLST) in 2018, withdrawal of life-sustaining treatment was not legally protected. The proportion of obtaining a Do-Not-Resuscitate order (DNR) from terminal stage patients had remained low and the DNR order was obtained too late in that 82.6% of patients died within 0–3 days after signing their DNR orders in 2006 [6]. Overall, in tertiary hospitals in Korea, adequate palliative care was not offered, referrals to hospices often failed, and there was no legal process to guarantee the end-of-life care that patients wanted.

Caring for critically and terminally ill patients is often challenging for physicians and nurses. In facing life-or-death matters, healthcare professionals hold moral beliefs in determining what is good for the patient. However, they simultaneously encounter various contradictory values and legal restrictions, and the practices are done customarily [7, 8]. Moral distress occurs when one knows the right thing to do but institutional constraints make it nearly impossible to pursue the right course of action [9]. Since the definition of Jameton, previous studies have explored the major elements and negative effects of moral distress, and discussed how to measure the severity of moral distress and how to reduce the occurrence of moral distress in clinical settings [10–13].

With a few exceptions [14], the literature focuses on a homogeneous group of healthcare professionals, such as nurses providing geriatric care, physician trainees, the intensive care unit (ICU) staff, and staff in the pediatric departments [11, 15–17]. Yet, a large-scale hospital is a setting where healthcare professionals in diverse departments work together with respective care goals and duty priorities under the complex interrelations of legal regulations and internal rules. Depending on their occupation (e.g., doctor or nurse) [18, 19], level of experience [11, 16, 20], and specialty (e.g., ICU or emergency medicine) [21, 22], healthcare professionals may encounter considerably different issues to which they respond in diverse ways [14]. In this

context, institutional constraints are highly likely to preclude healthcare professionals from pursuing what they believe to be morally appropriate for patient care [23].

The aim of this study was to explore moral distress in the context of care for critically and terminally ill patients in tertiary hospitals by analyzing the experiences of physicians and nurses from various departments. In doing so, the present study sought to identify what healthcare professionals believe to be right, what institutional constraints hamper the values and beliefs they wish to pursue, and what consequences the resulting moral distress has on them.

## Method

### Design

Semi-structured in-depth interviews were conducted to explore healthcare professionals' experiences of moral distress related to caring for critically and terminally ill patients.

### Recruitment

Physicians and nurses who had previously provided care for critically and terminally ill patients and had experienced ethical challenges were recruited from two tertiary university hospitals in South Korea. These two hospitals had palliative care team and ethics consultation services. To obtain a diverse array of views and experiences, this study used purposive maximum variation sampling [24, 25]. A total of 22 participants were recruited through advertisements posted on bulletin boards in both hospitals and an online bulletin board in one of the hospitals. None of the participants were related to the research team. All participants were informed about the purpose and methods of the study as well as the constitution of the research team. Written consent was obtained prior to their participation.

All the interviews began with the question, "Could you describe the personal experience that may have prompted you to participate in our study upon seeing the advertisement?" By starting with such an episode, participants were able to freely explore the situations that had caused moral distress, the burdens they had experienced, how they had tried to resolve the incidents, and the root causes of the episodes. The interviews were recorded with the participants' approval and transcribed. All identifying data were deleted from the transcripts. Theoretical saturation was reached when no new issues were addressed in the interviews. Consequently, recruitment was closed. One member of the research team (JK), who was an experienced ethnographic researcher and a medical anthropology doctoral candidate, conducted all interviews.

### Data collection

In the research preparation stage, a list of questions was prepared to guide the semi-structured interviews (Table 1).

**Table 1. Semi-structured interview questions.**

| |
|---|
| 1. Have you experienced ethical concerns or moral conflicts while caring for a critically and terminally ill patient? |
| 1–1. What did you want to do for the patient? |
| 1–2. Why did your ethical concerns or moral conflicts occur? |
| 2. Did you feel any physical or psychological burden because of these moral conflicts? |
| 3. How did you try to solve the problem? |
| 4. What advice would you like to give someone who is experiencing a similar problem? |

## Analysis

The transcripts were analyzed according to grounded theory [26] using MAXQDA. The grounded theory approach was adopted as it is a useful analytic tool for research aiming to uncover problems faced by participants within a particular context. There is little research regarding how Korean health practitioners relate to moral distress when caring for critically and terminally ill patients in acute care institutions. Following a data-driven strategy [25], in the stage of open coding, two members of the research team (EC and JK) independently created the core categories and key themes with memos that emerged while they were iteratively reading the transcripts. The categories and themes that were addressed in previous interviews were redefined in subsequent interviews and informed further data collection. In the axical coding stage, EC and JK linked the emerged core categories and key themes to understand their causal relationships and subsequently developed a codebook. Another researcher (HYP) examined the transcripts and the codebook together. The drafting of the codebook was supervised by the entire research team and an external expert in nursing-related qualitative research in the selective coding stage. After three rounds of supervision, the final comprehensive code structure that explained the relationships between the core categories and themes was approved. Through multiple rounds, all members (seven women and one man) from diverse positions (medical student, doctoral candidate, and professor) and various backgrounds (medical anthropology, medicine history, nursing, palliative care, and oncology) examined the interview data in a reflexive manner, and this process contributed to minimizing biased interpretation.

## Ethics

The ethics committees of the Seoul National University Hospital and the Seoul National University Bundang Hospital approved the study protocol and materials in December 2016 (No. 1612-071-813). This study was conducted in accordance with the Declaration of Helsinki.

## Results

Nine physicians and 13 nurses participated in the study from February to June 2017. All interviews were conducted by a single researcher (JK) and lasted between 60 and 90 minutes each. The participants' demographic characteristics are displayed in Table 2.

Among the 22 participants, 21 reported the experiences that conformed to the definition of moral distress, that is, they knew what was right for the patients but were unable to pursue their values and beliefs due to institutional constraints. One nurse working in the ICU addressed emotional distress caused by caring for the dying patients rather than moral distress, and this data was not considered in this study. The data analysis identified what the participants valued for the patients (Table 3).

### Minimizing meaningless intervention

Once a patient was diagnosed to be at the terminal stage, participants valued minimizing tests or treatments that caused pain and discomfort to the patient. Participants did not see any benefit that could be gained from the interventions because these interventions did not contribute to the patient's recovery or improve their quality of life. The scope of the meaningless interventions that participants addressed included not only aggressive life-sustaining treatments but also ordinary tests, such as arterial blood collection, and treatments (e.g., anti-cancer treatment).

**Table 2. Demographic characteristics of the study participants ($n$ = 22).**

|  | $N$ | % |
|---|---|---|
| **Sex** |  |  |
| Female | 18 | 81.81 |
| Male | 4 | 18.18 |
| **Age (years)** |  |  |
| 19–29 | 6 | 27.27 |
| 30–39 | 12 | 54.54 |
| 40–49 | 3 | 13.63 |
| 50–59 | 1 | 4.54 |
| **Profession** |  |  |
| Doctor | 9 | 40.9 |
| Nurse | 13 | 59.1 |
| **Years of employment** |  |  |
| < 5 | 8 | 36.36 |
| $5 \leq n < 10$ | 7 | 31.81 |
| $10 \leq n < 15$ | 5 | 22.72 |
| $\geq 15$ | 2 | 9.09 |
| **(Doctors) Specialty** |  |  |
| Emergency medicine | 1 |  |
| Pediatrics | 2 |  |
| Neurology | 3 |  |
| Intensive care medicine | 2 |  |
| Family medicine | 1 |  |
| **(Doctors) Level** |  |  |
| Resident | 4 | 44.44 |
| Fellow | 1 | 11.11 |
| Professor | 4 | 44.44 |
| **(Nurses) Department** |  |  |
| Intensive care units | 4 | 30.76 |
| Wards | 4 | 30.76 |
| Outpatient departments | 5 | 38.46 |

"I needed to do a CO2 test to an elderly patient with pneumonia and stroke to see how bad his condition was. I collected arterial blood for it, and he was basically supposed to be stabbed in his arteries four times a day. . . I was like, 'Do I have to do this to this dying man? Is it really for the good? I can't see anything but harm resulting from it.' But I was an intern and interns just do what is supposed to be done." (ID 12. Resident, Family Medicine)

Although participants regarded these interventions as "treatments that only makes it harder for the patient," they had to initiate or continue these interventions on their patients because

**Table 3. What participants valued for the patients.**

| |
|---|
| Minimizing meaningless intervention at the terminal stage |
| Letting patients know of their bad prognosis |
| Saving lives |
| Necessity of palliative care |
| Care with compassion |

of the hierarchy among healthcare professionals. Participants, as interns, residents, or nurses, could not challenge the orders of their supervisors. Nurses stated:

> "I am not in the position to make any decision. It is the professors who make decisions. It is hard for me to say something when my patients ask about my opinion." (ID 34. Nurse, Internal Medicine)

> "I know this is not right and I know patients have hard times, but at times, I lose my motivation because I think 'Oh, this seems meaningless,' while doing my job. (ID 28. Nurse, EICU)

Residents and nurses were not the only participants who had to perform what they regarded as meaningless interventions. Attending physicians also reported that they considered the patients' family members as the final decision-makers over patients themselves. All participants thought that family members take the ultimate responsibility as they would be able to file a lawsuit against the healthcare professionals.

> "The opinions of the family members are important because the patients in ICUs are unconscious. Even if you have written a DNR, have expressed what you would want at the end of life, like, 'I don't want respirators, I don't want CPR (cardiopulmonary resuscitation), I don't want dialysis,' your family members do not agree and ask me to do something, then I am supposed to do that." (ID 16. Nurse, EICU)

> "If a patient's relatives assert that they can handle it, afford the treatment cost, and hire paid caregivers, we do it [hopeless life-sustaining treatment]. Even though the patient expressed her will not to receive life-sustaining treatment, we can't help but do it because there is no legal protection for us." (ID 21. Physician, Intensive Care Unit)

**Letting patients know of the poor prognosis.** Participants believed that the patients should know of the poor prognosis in advance so that the patients and their family members can prepare for their death, and accordingly consider decisions such as spending time with the family, enjoying their favorite food, having a chance to leave a will, and deciding whether to receive life-sustaining treatment. However, nurses were not allowed to talk to their patients about poor prognosis before the attending doctor and the family caregivers of the patient brought up the topic with the patients.

> "Sometimes, patients call us and ask about their prognosis. I can't say anything because the professor must have an idea but I don't know what it would be, and my patients want to hear my opinion. I'm in a very awkward position." (ID 1. Nurse, Outpatient Department)

Combined with their position on minimizing meaningless interventions, obtaining a DNR order from a patient at the terminal stage was an important means to initiate the end-of-life care plan discussion with the patient in advance. Simultaneously, however, all participants shared the idea that obtaining a DNR was regarded as giving up on the patient by healthcare professionals, patients, and their family members. Theoretically, a DNR order means no cardiopulmonary resuscitation, but in the acute care setting, all participants felt that a patient was abandoned once they signed their DNR. Consequently, residents whose practices were under evaluation were worried that obtaining a DNR order would be perceived as negligence even though they believed it was necessary for the patient to die with dignity. One resident said:

"I feel that obtaining a DNR order means no more treatment. If I try to get one from a patient, it will look like I don't want to care for them. I was told by my senior that an internal medicine professor admonished residents for obtaining DNR orders, saying that they did it because they didn't want to do their job. That means that professors regard a DNR order as abstaining from further treatment." (ID 12. Resident, Family Medicine)

In addition, the prevailing clinical culture of avoiding the discussion of death frustrated participants. They thought the current timeframe for obtaining a DNR order is too late to give the patients time to prepare for their death. However, in circumstances where no one is expected to talk about end-of-life plans until "the death rattle appears," participants could not break this tacit rule. Some participants shared fear with the patients' relatives that the poor prognosis might frustrate the patient and hasten the patient's death.

"When a doctor brings up the topic to the patient's family members, they also put off the conversation, like 'We need time.' All the people know what is going to happen but wait until it happens." (ID 19. Nurse, Oncology Ward)

"If you try to get a DNR from a patient with clear consciousness, you might frustrate the patient." (ID 34. Nurse, Internal Medicine)

**Saving lives.** Some participants reported their experiences of situations where they wanted to fulfill the primary goal of medicine, which is to save lives, but could not do so despite their medical judgment or were unsure if they did the right thing. First, when a patient goes through an event at the terminal stage, physicians and nurses may ascertain that the patient can survive for a few more weeks and months through some interventions. However, the family members of the patient may regard this intervention as prolonging the patient's suffering.

"An infant got brain damage and was expected to be bedridden. The parents were doctors. Maybe they knew the prognosis too well. They wanted to give up on the baby. There was nothing we could do. When the baby was discharged, she was okay except for the brain damage, and she died at home." (ID 15. Nurse, NICU)

Second, participants from the neurology and pediatrics departments addressed the episodes in which the practices they employed to save the patients led to a bedridden state. According to the participants, with the advancement in medical knowledge and technologies, the survival rates of critically ill patients have increased, but in many cases, the "saved" patients have to depend on other people's care for the rest of their lives. Participants acknowledged the situations where public resources to support these patients are not enough, and family caregivers are in charge of round-the-clock care for the patients.

"In the case of child patients, they live in that [bedridden] state from when they are a few months old. Their mothers have to take care of them all the time, without any promise of getting better... If your parents or grandparents are in the bedridden state, you are subsidized by the state and there are some institutions you can take your parents to. But if children are bedridden, it is only their mothers who take all of the care burden. I have some child patients who have survived for 10 and 20 years. Their mothers are getting older and depressed because they don't have their own lives but are taking care of the sick children." (ID 41. Physician, Pediatrics)

As participants had observed the family caregivers' physical and psychological burnout and financial troubles, they were sometimes not sure if trying to save their patients was the best measure for everyone.

"Sometimes I think if my job is supposed to let children patients go [die], should I do everything for my child patients when CPR is needed or should I pretend to do something and just send the child to God and take the burden off the mother?" (ID 40. Physician, Pediatrics)

**Necessity of palliative care.** Participants acknowledged the necessity of special care for patients at the terminal stage, including pain management, comfort care, psychological care, and spiritual care. However, tertiary hospitals in Korea are not sufficiently prepared to provide palliative care services to all patients. Consequently, physicians and nurses who have not specialized in end-of-life care are put in charge of dying patients. For instance, a nurse working in an oncology ward said:

"I get irritated when physicians order pain control medications as if they are mechanically following the textbook. I wonder if they even think about how severe the pain is that the terminal cancer patients are experiencing. They just follow the textbook stating, 'maintain four-hour interval, two-hour interval, dose according to body weight.' . . . I got a patient, and he was like, 'Please save me, please give me the painkillers,' even as I *was* giving it to him. He was terrified about the pain that will come next time. So, I hang more painkillers next to the IV poll and assure them that I will give it to them whenever they can't stand the pain. Patients remember how severe the pain is and they are very anxious. Time interval does not matter. How on earth can you stand cancer pain for four hours?" (ID 20. Nurse, Oncology Ward)

While this nurse addressed the lack of personnel resource in palliative medicine, another nurse working in the oncology ward mentioned about another issue in the ward. The recruitment hospital was equipped with a single-bed room that was reserved for a dying patient (whose death was expected within two to three days) in the oncology ward, but the nurse taking care of the dying patient in the single-bed room felt powerless and frustrated because she had no further end-of-life care options besides offering the room.

"We feel stressful, too. I have nothing that I can do for them [the patients in the single-bed room], but they are conscious, and the family members are there. I don't know what to say to them. It is awkward to ask them, "Are you ok?" isn't it?" (ID 19, Nurse, Oncology Ward)

**Care with compassion.** One theme that frequently emerged was the value of care with compassion and the frustration that it wrought. Participants weighed providing sincere care to their terminally and critically ill patients and their family members. However, the healthcare professionals working in tertiary hospitals reported pressure from hospital administrators to save time and beds.

"The problem is that these patients tend to stay for a long time, which generates 'a traffic jam.' The bedrooms in this hospital are not for them. You know, we have patients who got a heart surgery and *they* need the bedrooms. If you are taking care of the terminally-ill patient for months, then it would cause a huge problem to all people here." (ID 40. Physician, Pediatrics)

According to the acute care logic, patients in the terminal stage are supposed to be moved to long-term care institutions so that patients with acute problems can be treated. The attending physicians, who had the responsibility of allocating the limited space to patients, felt that they were caught in between the acute care logic and their wish to provide quality care for terminally-ill patients.

In terms of pressure to save time, physicians reported that they had only five minutes to discuss the end-of-life care plan, including withdrawal of life-sustaining treatment; the workload of the nurses was too heavy for them to provide humane care to their patients. For example, one nurse said:

> "I don't have time to talk with a patient or their family members during working hours. I don't even have time to pee for 12 hours." (ID 34. Nurse, Internal Medicine)

In the work environment where the participants were sometimes unable to fulfill their own basic needs, they viewed their patients only as "work." Detachment from their patients functioned as a coping strategy to deal with the emotional distress and heavy workload, but this strategy did not resolve the fundamental problem in that it made them feel guilty.

> "Sometimes I feel callous, and this feeling causes internal conflict in me. You can't respond as much as people expect you to when they are sick and dying. You can't show your sympathy or console them and just say, 'That's how it is,' and your response upsets people." (ID 3. Nurse, Outpatient Department)

Since obtaining a DNR order, as mentioned above, means cessation of treatment, nurses experiencing intensive workload felt less burdened once a patient had signed one. This feeling, however, conflicted with their professional ethics, and they experienced guilt for feeling relieved by a patient's death.

> "You are busy working and you take care of your job mechanically. If you get the DNR form from a dying patient, you [the nurses and doctors] feel relieved because it is agreed by all that this patient is dying. Even if the patient's pulse suddenly drops, it is okay because you have gotten the consent from the family members and it's okay to see her die. . . Of course, I'm a little upset, but I'm a little relaxed. I feel less pressure." (ID 27. Nurse, Hematology)

**Outcomes of moral distress.** Participants reported particular emotional responses and behavioral strategies used to handle problematic situations (Table 4).

**Table 4. Consequences of moral distress.**

| |
| --- |
| Trying a family-centered round |
| Helplessness |
| Stress |
| Lack of enthusiasm |
| Guilt |
| Depression |
| Skepticism |
| Detachment from patients |
| Quitting jobs or changing specialty |

One participant (Intensivist, Professor) reported her attempt to minimize moral distress situations in ICUs. Since withdrawing and withholding life-sustaining treatment is a crucial issue, and agreement among healthcare professionals, patients, and patients' family members is the key to solving this without fear of lawsuit, she and her colleagues were trying a new mode of consultation: a family-centered ICU, which satisfied both the healthcare professionals and the patients' family members, and contributed to reducing potential cases that could cause moral distress.

However, most of the participants did not have the resources or the power to challenge institutional constraints. Residents and nurses reported having no reliable sources of advice and approaching the hospital ethics committee was not regarded as an option. Stress, lack of enthusiasm, guilt, depression, and skepticism were reported, with some participants deciding to be detached from their patients for self-protection.

> "I said to my colleague, 'I think I will go to hell.'" (ID 11. Resident, ER)

> "I am still looking for a good answer. But if someone asks my advice, I would say, 'Do not work here for a long time.'" (ID 15. Nurse, NICU)

Three participants quit their job or changed their specialty to get away from the situations that generated conflicts between what they wanted to pursue and what hampered the same. For instance, a participant explained the reason for changing her specialty:

> "I was like, 'Should I do this [unnecessary exam] to dying patients?' and I wanted to do something that prevents people contracting diseases, and I changed my way from internal medicine to family medicine." (ID11. Resident, Family Medicine)

## Discussion

The findings of this study are largely consistent with the themes and root causes reported in previous studies that were conducted in other countries and in non-acute care settings [14, 21, 27–31]. The values that participants in this study stated are in line with the studies conducted in nursing homes [10] and acute hospital settings in the UK [32], and among physician trainees in the USA [11]. Akin to many studies [10, 33, 34], nurses in this study have addressed the difficulties originating from the hierarchy among healthcare practitioners and the proximity with their patients. This supports the notion that nurses may be the first to recognize the patients' worsening state but their lack of influence over critically and terminally ill patients causes moral distress. The physician trainees' report regarding powerlessness while providing "futile" treatment is also aligned with previous studies [11].

The results of this study reconfirm the unique qualities of acute care settings that have been addressed in previous literature. In the acute care settings where patient groups mix and the work pace is hectic, care of dying patients is given a lower priority [33, 34]. This work environment hampers the healthcare practitioners' efforts in providing sincere and humanistic care for critically and terminally ill patients and their families [7]. It is also seen that hospitals tend to pay less attention to palliative care services [33, 35, 36], as participants in this study wished to have more staff and facility resources for palliative and hospice care.

Even though the necessity of palliative care in acute care settings has been highlighted [37], in these medical institutions, where the palliative philosophy conflicts with biomedical culture that is heavily oriented toward cure and recovery, the initiation of palliative care tends to be seen as giving up on the patient [32], and consequently, end-of-life care discussions start too late [32]. Palliative care is still practiced according to the transitional model of care that sharply

divides curative from palliative care and is inappropriately conducted in a fragmented way [38]. Further, this study also found that moral distress negatively affects healthcare professionals' job retention [14, 28, 39, 40], to the extent that some participants considered leaving their departments or had changed their specialties.

Notably, two themes which have been rarely addressed in previous literature emerged in this study. First, this study found saving lives as one of the values in relation to end-of-life care. This theme was addressed among challenging cases of pediatrics and neurology departments survivors with significant disability. The doctors and nurses understood the lack of social support for these survivors and caregivers as a root cause of moral distress. This finding indicates that in order to reduce experiences of moral distress, public care infrastructure should be considered for survivors and their caregivers after medical interventions save them at the crossroads of life and death.

Second, it is also noteworthy that aggressive treatment was not the only issue for the physicians and nurses in acute care settings. Moral distress in end-of-life care tended to be discussed in relation to the meaninglessness of life-sustaining treatments for terminally ill patients [11, 28, 39, 41]. However, participants regarded the lack of comfort care after the patients entered the terminal stage and difficulties in transition from treatment for cure to palliative and hospice care, too, as crucial elements of moral distress. This finding suggests an apparent need for palliative care specialists, both for patients and medical staff, even in large-scale acute care settings, and that not only the prevention of futile life-sustaining treatment but also the facilitation of comfort care is an urgent needs.

## Conclusion

Physicians and nurses working in acute care settings, such as tertiary hospitals, take care of critically and terminally ill patients and are involved in various stages of care, ranging from diagnosis of life-threatening diseases to care for the dying. Regardless of the multifaceted stages and situations of dying and death, and regardless of the large number of hospital deaths, tertiary hospitals in South Korea do not provide enough palliative care services to patients. The number of hospices remain insufficient to cover terminally ill patients. Given this context, the present study had aimed to identify the moral distress experienced by healthcare professionals working in the two large-scale tertiary hospitals, in relation to care for critically and terminally ill patients. Among the 22 participants, 21 reported experiences of moral distress. What the participants believed to be right for the patients included minimizing meaningless interventions during the terminal stage, letting patients know of their poor prognosis, saving lives, offering palliative care, and providing care with compassion.

However, family dominance, hierarchy, the clinical culture of avoiding the discussion of death, the lack of support for patients who survived, and the intensive workload challenged and frustrated what the participants were pursuing. As a result, the participants experienced stress, lack of enthusiasm, guilt, depression, and skepticism. The in-depth interviews in this study allowed the participants to freely and honestly explore what they had undergone, thought, and felt regarding critical and terminal care in the clinical field. Given that a large number of the previous studies have been conducted in North America and the UK, and in nursing homes and hospices, this study contributes to the dialogue by adding data collected from two tertiary hospitals in South Korea.

This study had some limitations. First, the experiences of the physicians and nurses that emerged from the interviews may have been influenced by cultural and contextual factors specific to the two institutions included in the study. Second, as only practitioners who had experienced moral distress volunteered to participate in this study, the cases reported here may not

be representative of other medical professionals. Further studies are needed to compare moral distress between institutions and countries.

## Acknowledgments

We would like to thank all the participants for their time and openness in discussing this issue.

## Author Contributions

**Conceptualization:** Jiyeon Kang, Eun Kyung Choi, Hye Yoon Park.

**Data curation:** Jiyeon Kang, Eun Kyung Choi, Grace S. Ahn, Hye Youn Park, Hye Yoon Park.

**Formal analysis:** Jiyeon Kang, Eun Kyung Choi, Grace S. Ahn, Hye Yoon Park.

**Funding acquisition:** Hye Yoon Park.

**Investigation:** Jiyeon Kang.

**Methodology:** Jiyeon Kang, Eun Kyung Choi, Hye Youn Park, Hye Yoon Park.

**Project administration:** Jiyeon Kang, Hye Youn Park, Hye Yoon Park.

**Resources:** Jiyeon Kang, Eun Kyung Choi, Hye Yoon Park.

**Software:** Jiyeon Kang, Eun Kyung Choi.

**Supervision:** Jiyeon Kang, Eun Kyung Choi, Minjeong Seo, Hye Youn Park, Jinui Hong, Min Sun Kim, Bhumsuk Keam, Hye Yoon Park.

**Validation:** Jiyeon Kang, Eun Kyung Choi, Minjeong Seo, Jinui Hong, Min Sun Kim, Bhumsuk Keam, Hye Yoon Park.

**Visualization:** Hye Yoon Park.

**Writing – original draft:** Jiyeon Kang, Eun Kyung Choi, Grace S. Ahn, Hye Yoon Park.

**Writing – review & editing:** Jiyeon Kang, Eun Kyung Choi, Minjeong Seo, Grace S. Ahn, Jinui Hong, Min Sun Kim, Bhumsuk Keam, Hye Yoon Park.

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
