## [Decision Letter · Decision Letter 0]

19 Nov 2020

PONE-D-20-27654

Imbalanced end-of-life care and moral distress in tertiary hospitals:

A qualitative study

PLOS ONE

Dear Dr. Park,

Thank you for submitting your manuscript to PLOS ONE. After careful consideration, we feel that it has merit but does not fully meet PLOS ONE’s publication criteria as it currently stands. Therefore, we invite you to submit a revised version of the manuscript that addresses the points raised during the review process.

While all reviewers acknowledge the importance of this study, they have also proposals for substantial revisions both at the conceptual and methodological level. I kindly ask you to evaluate everything in detail and to revise the paper accordingly. Please note that the paper will be re-assessed and editorial decision will be made on the basis of the second round of reviews. 

We look forward to receiving your revised manuscript.

Kind regards,

Sara Rubinelli

Academic Editor

PLOS ONE

Journal Requirements:

Reviewers' comments:

Reviewer's Responses to Questions

**Comments to the Author**

1. Is the manuscript technically sound, and do the data support the conclusions?

Reviewer #1: Yes

Reviewer #2: Partly

Reviewer #3: Partly

2. Has the statistical analysis been performed appropriately and rigorously? 

Reviewer #1: N/A

Reviewer #2: N/A

Reviewer #3: N/A

3. Have the authors made all data underlying the findings in their manuscript fully available?

Reviewer #1: Yes

Reviewer #2: No

Reviewer #3: Yes

4. Is the manuscript presented in an intelligible fashion and written in standard English?

Reviewer #1: Yes

Reviewer #2: Yes

Reviewer #3: Yes

5. Review Comments to the Author

Reviewer #1: Thank you for the opportunity to review this manuscript on the increasingly important topic of moral distress among clinicians.

The authors have presented the data well and it is well written especially as English is not their first language. Additionally, it is important to have data from different countries as the embedding of palliative care into clinical care differs in many settings.

The authors have highlighted important cultural differences in particular in both clinician and family views of the role of palliative care. It is disappointing that it is seen as "giving up" and that "nothing more can be done". This indicates the need for education both at a clinical and community level which the authors highlight in the Discussion.

The themes are well described however, the findings needed to be structured more around the Semi-structured interview questions. The responses to these questions do not appear in the manuscript at all. The paper is more focussed on the failings of providing end of life care rather than the moral distress that this causes.

I would also like to see a definition of critical incident techniques which was the theoretical basis of the analysis.

The Abstract needs to be stronger to reflect the findings and the conclusion needs to be more aligned with the question of moral distress - which is the theme of the study - hence the statement from the manuscript that "Medical practitioners experienced moral distress either when providing futile life-sustaining treatment or failing to provide comfort care”. This I see as the key message rather that a statement that the study demonstrated that end-of-life care is neglected in acute care settings or a listing of the ways end of life care has failed.

The study brings important insights, however it needs to be more focussed on the main question of the study.

Reviewer #2: Thank you for the opportunity to review your manuscript, ‘Imbalanced end-of-life care and moral distress in tertiary hospitals: A qualitative study’.

The topic is a worthy one and the study shines a light on the experiences of physicians and nurses in Korea in end of life care that causes them moral distress.

However, the manuscript requires attention in the following areas before it is ready for publication.

1. Title: should include participants, as it is not clear whose moral distress was examined.

2. Abstract:

a. Suggest you do not use the term ‘medical practitioners’ to describe the physician and nurse participants, as this term signifies physicians in many parts of the world. Suggest the terms ‘health professionals’ or ‘clinicians’ would be more universally correct.

b. The assertion that the study demonstrates that end of life care is neglected in Korean hospitals is too strong/too great a leap for a qualitative study of the perspectives of physicians and nurses. Please revise this statement so that it is more in keeping with the methods/objectives of your investigation.

3. Introduction:

a. Please explicitly define end of life care, as this is your key concept, and it can be interpreted differently. For example, in Australia end of life care is defined as that which aims to optimise quality of life for persons likely to die in the next 12 months by addressing suffering, promoting dignity and function, respecting their needs and preferences, and supporting their family, including into bereavement. Is this what you are referring to by end of life care?

b. Suggest you revise use of the term ‘a good death’ (page 3), as this concept differs from end of life care, and furthermore it is a loaded and variously interpreted term.

c. Suggest use of less euphemistic language than ‘meet their demise there’ (paragraph 2, page 3) e.g. ‘die there’.

d. Please provide a reference/s to substantiate the low level of satisfaction with end of life care in particular regions of the world (page 3).

e. The introduction should include information about the end of life care needs, services and health care policy in Korea, to communicate to the reader the context in which the study was conducted.

f. The last paragraph of the introduction (the aim of the study) contains two different and seemingly dissonant objectives. The first was to examine physicians’ and nurses’ subjective experiences of moral distress related to end of life care. The second was to identify the organisation and systemic barriers to EOL in the hospitals, which strikes me as being more suited to an objective assessment, and too great a leap from the first objective. Please re-consider the scope of examination of the topic that was possible using qualitative methods.

4. Methods:

a. The manuscript states that the COREQ reporting guideline informed the study methods. This statement is problematic for two reasons. Firstly, reporting guidelines are to guide study reporting, not study methods. Secondly, the study has not been reported according to the COREQ, as several items are missing. Rather than pointing each of these out individually, my suggestion is that you ensure that all items of the COREQ are reported in your revised manuscript.

b. Table 1 belongs in the results section, not the methods, as does the information about duration of the interviews.

c. Critical incident technique has a particular analytic method, whereas you used a grounded theory analysis approach. Currently, the reporting suggests an unsound hybrid qualitative approach, especially as the findings are not reported in line with critical incident technique. Please revise this section to justify why you used the hybrid approach.

d. Analysis: last line - what specific type of bias did you minimise? Also, because you have not reported your positions, backgrounds and reflexivity process (which the COREQ calls for), it is difficult for the reader to be convinced that you fully addressed the minimisation of researcher bias.

5. Findings:

a. An introductory paragraph, including the characteristics of participants and interview, and summary of the themes and sub-themes, is required. Also, I was left wondering what proportion of the sample experienced moral distress.

b. In reporting the findings, it is important not to over-generalise. For example, the statement that procedures and treatments were not provided in patients’ best interests (page 8) should be modified, as you are recounting health professionals’ subjective perspectives only.

c. The statement “Because obtaining a DNR order means cessation of treatment…” requires revision, as a DNR order means no cardiopulmonary resuscitation, not no treatment overall.

d. I believe that the lack of exemplar quotes in the reporting of the ‘Main contributing factors’ theme adversely affects the strength and trustworthiness of your interpretations. While they are available in the supplementary file, their absence in the narrative means that the findings are reported in a disjointed way. Please consider formatting the findings in a consistent way throughout, and provide exemplar quotes for each sub-theme.

6. Discussion:

a. The first paragraph of the discussion is clear and reflects the findings.

b. However, at times you introduced new findings in the discussion that were not earlier reported. For example, I can’t see anything that refers to patients’ pain in the findings, but it is referred to on page 13. Similarly, there were no data on participants wanting to leave their work, yet it is referred to on page 14. Only items reported in the findings should be discussed.

c. As per my earlier comments about there being too great a leap from investigating participants’ moral distress to interpretations about systems/macro level change, I suggest you rein in recommendations about macro-level transformation; or at least, provide more specific recommendations about required practice and system change.

d. Please provide a conclusion.

7. References: A key missing reference is: ‘Re-defining moral distress: A systematic review and critical re-appraisal of the argument-based bioethics literature’ https://journals.sagepub.com/doi/abs/10.1177/1477750919886088

8. Figure: I found it strange that moral distress of health professionals was the end point, whereas really it was your starting point. Also, it suggested to me a focus on health professionals’ well-being, rather than patients or their families, who are the primary focus of end of life care. I suggest that some more thinking is required about how to best convey the findings of your study diagrammatically.

I hope these suggestion are helpful in your revisions and wish you all the best in your ongoing work to improve end of life care in Korean hospitals.

Reviewer #3: Thank you for the opportunity review this paper, which explores end of life care and moral distress in tertiary hospitals in Korea. The authors have interesting data and raise important points, however, overall there are a number of methodological issues with the paper and overall, I find it quite confusing.

I think that this paper needs a bit of re-focussing and I wonder if authors are trying to do too much by exploring moral distress and imbalance of services. Regarding the latter, this feels a little bit tagged on and I wonder whether this would be a stronger paper I it focussed upon causes of moral distress in S Korean acute settings – using organization of services aspects as discussion. As currently written, I think the introduction needs to more clearly contextualise the paper in terms of: known causes of moral distress in the context of palliative care, and description of the South Korean health system – it is only in the results that palliative and hospice services are referred to. Methods also need improved description and justification of approach.

This is interesting work in need of improvement ahead of being publishable work – I have commented in more detail on specific sections below.

Introduction

The concept of moral distress is interesting, but it would be useful to have some examples here – what would be examples of the ‘diverse and even contradictory ethical and cultural values, as well as legal restrictions’ in which clinicians works in S Korea?

The claim that “tertiary hospitals serve as the starting points of end-of-life care” requires justification. You must forgive my lack of contextual knowledge of the S Korea healthcare system, but it would be useful to have some description of what other services/alternatives are available to the acute setting i.e. we learn that hospice is available in the results, is the suggestion that some people go to acute care when they more appropriately be under hospice care? Also, examples are given of western countries – many of which will have hospital palliative care teams (even if satisfaction remains low) – are any such services available in S Korea?

Overall, I don’t feel that the introduction justifies the aims. It feels more like you are focussing upon factors which contribute to moral distress in acute hospitals? The additional aim regarding organization of care feels a little tagged on and without improved description of the healthcare system overall, difficult to understand.

Methods

Most importantly, I am not satisfied that the authors use Grounded Theory, which is not something which can simply be applied as a mode of analysis. GT applies far earlier than the analysis stage an would also require continuing analysis, used to inform further data collection. The analysis process which the authors describe sounds entirely suitable and of due rigor, but is not GT and requires different description.

As for the sampling strategy, authors state that ‘purposive maximum variation’ sampling, but also that participants were recruited through open advert. Please make clear how this approach allowed a diverse range of views and also provide comment on how the sample size was agreed and its representation of the workforce within acute hospitals across departments.

Thank you for including the interview questions – I think they look good but also echo that organization of care was not a key aim.

Results

First line states that ‘overtreatment and a lack of care emerged from the interviews as issues causing the participants’ moral distress’. I feel that it needs to be established that participants did experience moral distress before making this claim. Did participants identify as having moral distress in interviews? Was this part of an inclusion/exclusion criteria? It is noted as a limitation that ‘all participants had ‘moral distress’ though I am not satisfied that this has been established – the recruitment poster only refers to inviting people who had provided terminal care.

Findings on Slow Code are heartbreaking, but require setting up better in the introduction to achieve full impact. In what circumstances do legal issues arise here to cause ‘fear of lawsuits?’ RE: DNRs, please clarify what is meant by ‘cessation of treatment’ – I would expect that people would continue to receive usual care in this circumstance, but would not be resuscitated.

In ‘Lack of end-of-life care resources,’ it’s stated that palliative and hospice care are marginalised in the hospital system, but this is the first suggestion that these services may be available at all! Please clarify.

Euthanasia is mentioned within ‘main results’ but this is the first use of the word in the manuscript. Is euthanasia practiced in S Korea?

I feel that more discussion is needed regarding the patient profile within an acute hospital in S Korea. Continuing life sustaining treatment can be a barrier to appropriate care, but all kinds of patients will be present in an acute setting, many of whom for which life sustaining therapy would be appropriate. Please comment on this issue and the prevalence of people with terminal illness in this setting.

Discussion

The Discussion as currently written is quite vague and a little confusing. E.g. the sentence “if a doctor does not believe that he or she will be supported by the hospital system when a patient’s relatives file a lawsuit, it is hard for the doctor to pursue the patient’s best interests or wishes.” Is somewhat alarming and poorly explained.

I would suggest that this section needs re-writing to both review the causes of moral distress and make clear its consequences. Description included within this section re: available services in acute settings, should come much earlier (introduction) so that Discussion can follow here.

“The six contributing factors that participants addressed are consistent with the themes and root causes reported in previous studies” – what are the contributing factors and what do they contribute to? – please refer to these factors or reference the figure earlier.

6. PLOS authors have the option to publish the peer review history of their article (what does this mean?). If published, this will include your full peer review and any attached files.

Reviewer #1: No

Reviewer #2: **Yes: **Annmarie Hosie

Reviewer #3: **Yes: **Dr Joseph Clark

---

## [Author Response · Author response to Decision Letter 0]

21 Jan 2021

We all deeply appreciate your thorough and sincere reviews. All the comments helped us to improve the original manuscript. We have submitted Response to Reviews.

---

## [Decision Letter · Decision Letter 1]

1 Mar 2021

PONE-D-20-27654R1

Care for critically and terminally ill patients and moral distress of physicians and nurses in tertiary hospitals in South Korea: A qualitative study

PLOS ONE

Dear Dr. Park,

Thank you for submitting your manuscript to PLOS ONE. After careful consideration, we feel that it has merit but does not fully meet PLOS ONE’s publication criteria as it currently stands. Therefore, we invite you to submit a revised version of the manuscript that addresses the points raised during the review process.

Thanks for the revision of the manuscript, that has improved the first version submitted. Yet, one of the reviewers has still major comments that I kindly ask to carefully consider and address. Please note that the final decision will be based upon this further revision. 

We look forward to receiving your revised manuscript.

Kind regards,

Sara Rubinelli

Academic Editor

PLOS ONE

Reviewers' comments:

Reviewer's Responses to Questions

**Comments to the Author**

1. If the authors have adequately addressed your comments raised in a previous round of review and you feel that this manuscript is now acceptable for publication, you may indicate that here to bypass the “Comments to the Author” section, enter your conflict of interest statement in the “Confidential to Editor” section, and submit your "Accept" recommendation.

Reviewer #2: (No Response)

2. Is the manuscript technically sound, and do the data support the conclusions?

Reviewer #2: Partly

3. Has the statistical analysis been performed appropriately and rigorously? 

Reviewer #2: N/A

4. Have the authors made all data underlying the findings in their manuscript fully available?

Reviewer #2: No

5. Is the manuscript presented in an intelligible fashion and written in standard English?

Reviewer #2: Yes

6. Review Comments to the Author

Reviewer #2: Thank you for the opportunity to comment on your revised manuscript, 'Care for critically and terminally ill patients and moral distress of physicians and nurses in tertiary hospitals in South Korea: A qualitative study'.

The revised version is much more coherent and well integrated, and you have addressed most of the reviewers' comments well.

However, there are a few important areas still needing attention before the manuscript is ready for publication, as follows:

Title/abstract/aim: The phrasing of what you examined is imprecise in these parts of the report. Moral distress is experienced by persons in specific situations, so a clearer and simpler way of expressing what you examined is exactly as you expressed it in the study design section i.e. ''...'health professionals' experience of moral distress related to caring for critically and terminally ill patients'. My advice is that you use this better phrasing at all times throughout the study report.

Introduction: Bravo - this is much more informative and focused. However, I suggest a final refinement of the first paragraph by rounding up the percentages to whole numbers and tying in similar findings e.g. 49.9% of hospital death in the UK and 50% of hospital deaths in Australia is basically the same proportion and could be expressed as 'Half of people in the UK and Australia die in hospital compared to more than three-quarters (76%) of people in Korea (and btw, is that figure for South Korea only or all of Korea?). Also, given much of the previous research related to your topic was likely conducted in the US, what are the US stats?

Aim statement (pages 4-5): is the first statement your aim and the second statement your objectives? As presently this reads as two somewhat different aims. Suggest revise this section for improved clarity.

Table 1: The red text indicates you revised this information, yet the wording appears exactly the same as the original version. Please would you explain this point? (especially as interview questions can't be changed retrospectively)

Analytic approach: I share Reviewer 3's concern that you have not fully applied grounded theory. I can see that you have used elements of grounded theory, so my recommendation to address this issue is that you state you used a qualitative approach based on grounded theory, and then state exactly what the applied aspects were. It is also important to state the rationale for your choice of analytical method. Given there were existing data on moral distress in health professionals in relation to care of people likely to die, a grounded theory approach seems inappropriate, unless you choose to use it because there were little existing data on this topic in South Korea.

Findings:

- This section is now more focused and clear, but there are many author statements made without the addition of supporting quotes. For your study findings to be trustworthy, every assertion you make about what participants said should be backed up by an exemplar quote. By doing this, you are also allowing the reader to hear the participants' voices, which provides a more compelling narrative.

- The quote from the paediatric physician indicated to me that he/she was tempted to or in fact had dissembled (or more bluntly, been dishonest) with others in the team or even the family of the critically ill child about their clinical interventions. It also suggests (although this may not be what they meant) that they took an active role in hastening the death of the child, which is euthanasia, unethical and unlawful. However, the participant may simply have meant that they allowed (or wished to allow) that the child die a natural death through the removal of life sustaining treatment, which is ethical and lawful IF done transparently, with supportive care, and with the consent of those who are the primary decision makers for the child, which is almost always their parents/family. Given that you understand the context in which this quote sits, and the importance of distinguishing clinicians' intentions in their care of people at the end of life, please provide more explanation of what the participant meant, and explicitly acknowledge its implications in the discussion. For example, if you discerned that participants were not always honest/transparent in their clinical communication and actions, what was driving this troubling issue? I realise that addressing findings that do not align with ethical practice is challenging, but am sure that you will be able to do this well in a way that maintains respect and compassion for the participants while honoring what is owed to patients and their families.

- Similarly, in the quote from the nurse about pain relief, what exactly did he/she mean? Did he/she mean that the pain relief should be individualised to the needs of the person? As it is entirely reasonable and good practice to order medication at standard regular intervals in palliative care contexts, as this is based on half-life of the drug. There are other avenues for responding to pain that is not controlled by regular doses of analgesia e.g. PRN dosing with close monitoring. This is another example of ambiguity of the participants' meaning that requires more explanation.

Discussion:

- More specific examples/details of similar/different findings from other studies is required, as the only one mentioned was about job retention.

- Second paragraph, page 16: here again the focus is attempting to be too broad. What is unique and interesting about your study is that it was set in in South Korea and has specific implications for the health care system there. Please revise this section so that it becomes more appropriately focused on the context within which the study was conducted.

- Please state limitations of your study at the end of the discussion.

Alignment with COREQ: Items 4,5,6,7,8,25, 28,29 and 30 have not been adequately addressed. I strongly encourage that you do so to ensure your study reporting is of the highest quality.

Reference 1 has a punctuation error.

I hope these suggestions are helpful and I look forward to seeing the next version of the report.

7. PLOS authors have the option to publish the peer review history of their article (what does this mean?). If published, this will include your full peer review and any attached files.

Reviewer #2: **Yes: **Annmarie Hosie

---

## [Author Response · Author response to Decision Letter 1]

15 Apr 2021

We do appreciate your sincere and thorough feedbacks.

---

## [Decision Letter · Decision Letter 2]

18 Jun 2021

PONE-D-20-27654R2

Care for critically and terminally ill patients and moral distress of physicians and nurses in tertiary hospitals in South Korea: A qualitative study

PLOS ONE

Dear Dr. Park,

Thank you for submitting your manuscript to PLOS ONE. After careful consideration, we feel that it has merit but does not fully meet PLOS ONE’s publication criteria as it currently stands. Therefore, we invite you to submit a revised version of the manuscript that addresses the points raised during the review process.

The paper has much improved after the revision. Yet, there are still some aspects to consider as two reviewers have highlighted. Specifically to the comments of the last reviewer, it would be good to state clearly (e.g. in the implications of the study) how the study addresses previous literature and provides innovative insight. 

We look forward to receiving your revised manuscript.

Kind regards,

Sara Rubinelli

Academic Editor

PLOS ONE

Journal Requirements:

Reviewers' comments:

Reviewer's Responses to Questions

**Comments to the Author**

1. If the authors have adequately addressed your comments raised in a previous round of review and you feel that this manuscript is now acceptable for publication, you may indicate that here to bypass the “Comments to the Author” section, enter your conflict of interest statement in the “Confidential to Editor” section, and submit your "Accept" recommendation.

Reviewer #2: All comments have been addressed

Reviewer #4: All comments have been addressed

Reviewer #5: All comments have been addressed

2. Is the manuscript technically sound, and do the data support the conclusions?

Reviewer #2: Yes

Reviewer #4: Yes

Reviewer #5: Partly

3. Has the statistical analysis been performed appropriately and rigorously? 

Reviewer #2: N/A

Reviewer #4: N/A

Reviewer #5: No

4. Have the authors made all data underlying the findings in their manuscript fully available?

Reviewer #2: (No Response)

Reviewer #4: Yes

Reviewer #5: Yes

5. Is the manuscript presented in an intelligible fashion and written in standard English?

Reviewer #2: Yes

Reviewer #4: Yes

Reviewer #5: Yes

6. Review Comments to the Author

Reviewer #2: I congratulate the authors on a very well written and deeply moving account of the findings of their qualitative study, and wish them well in your ongoing work in this important area of patient care.

Reviewer #4: Thr revised artcile has addressed the outstanding concerns but would beneifit from a structured abstract with a more considered conclusion.

Reviewer #5: In this paper, a qualitative study describes the ethical conflicts of medical staff in Korean acute care hospitals. Qualitative research has been conducted by taking over the appropriate method, but I have some problems about the interpretation of the results.

1. The results of the interview are the five things shown in Table 3, but the conclusion is that 'As a result, the participants went through stress, lack of enthusiasm, guilt, depression, and skepticism.' are written and there is a leap in interpretation.

2. This study reveals much the same results as similar previous studies, as the authors have stated in their discussions. Therefore, I cannot find the strength or novelty of this research.I suggest that the author examine the results more deeply and consider the categorized list.

7. PLOS authors have the option to publish the peer review history of their article (what does this mean?). If published, this will include your full peer review and any attached files.

Reviewer #2: **Yes: **Annmarie Hosie

Reviewer #4: **Yes: **Dr Jonathan Koffman

Reviewer #5: No

---

## [Author Response · Author response to Decision Letter 2]

1 Aug 2021

We appreciate the thorough and kind reviews. Your comments truly helped us to improve the manuscript. We have included Response to Reviewers file in this revision.

---

## [Decision Letter · Decision Letter 3]

9 Nov 2021

Care for critically and terminally ill patients and moral distress of physicians and nurses in tertiary hospitals in South Korea: A qualitative study

PONE-D-20-27654R3

Dear Dr. Park,

We’re pleased to inform you that your manuscript has been judged scientifically suitable for publication and will be formally accepted for publication once it meets all outstanding technical requirements.

Kind regards,

Sara Rubinelli

Academic Editor

PLOS ONE

Additional Editor Comments (optional):

Reviewers' comments:

Reviewer's Responses to Questions

**Comments to the Author**

1. If the authors have adequately addressed your comments raised in a previous round of review and you feel that this manuscript is now acceptable for publication, you may indicate that here to bypass the “Comments to the Author” section, enter your conflict of interest statement in the “Confidential to Editor” section, and submit your "Accept" recommendation.

Reviewer #2: All comments have been addressed

2. Is the manuscript technically sound, and do the data support the conclusions?

Reviewer #2: Yes

3. Has the statistical analysis been performed appropriately and rigorously? 

Reviewer #2: N/A

4. Have the authors made all data underlying the findings in their manuscript fully available?

Reviewer #2: No

5. Is the manuscript presented in an intelligible fashion and written in standard English?

Reviewer #2: Yes

6. Review Comments to the Author

Reviewer #2: Again I congratulate the authors on their well-written study report. I have no further suggestions to make.

7. PLOS authors have the option to publish the peer review history of their article (what does this mean?). If published, this will include your full peer review and any attached files.

Reviewer #2: **Yes: **Annmarie Hosie

---

## [Editor Report · Acceptance letter]

7 Dec 2021

PONE-D-20-27654R3 

Care for critically and terminally ill patients and moral distress of physicians and nurses in tertiary hospitals in South Korea: A qualitative study 

Dear Dr. Park:

I'm pleased to inform you that your manuscript has been deemed suitable for publication in PLOS ONE. Congratulations! Your manuscript is now with our production department. 

Kind regards, 

on behalf of

Dr. Sara Rubinelli 

Academic Editor

PLOS ONE